# In Use Determination of Aerodynamic and Rolling Resistances of Heavy-Duty Vehicles

**Dimitrios Komnos** [1,2], **Stijn Broekaert** [3], **Theodoros Grigoratos** [3], **Leonidas Ntziachristos** [2] **and Georgios Fontaras** [3,*]

1   FINCONS Group, 20871 Vimercate, Italy; Dimitrios.Komnos@ext.ec.europa.eu
2   Mechanical Engineering, Aristotle University of Thessaloniki, 54124 Thessaloniki, Greece; leon@auth.gr
3   European Commission Joint Research, 21027 Ispra, Italy; stijn.broekaert@ec.europa.eu (S.B.);
    theodoros.grigoratos@ec.europa.eu (T.G.)
*   Correspondence: Georgios.Fontaras@ec.europa.eu

**Abstract:** A vehicle's air drag coefficient (Cd) and rolling resistance coefficient (*RRC*) have a significant impact on its fuel consumption. Consequently, these properties are required as input for the certification of the vehicle's fuel consumption and Carbon Dioxide emissions, regardless of whether the certification is done via simulation or chassis dyno testing. They can be determined through dedicated measurements, such as a drum test for the tire's rolling resistance coefficient and constant speed test (EU) or coast down test (US) for the body's air Cd. In this paper, a methodology that allows determining the vehicle's *Cd·A* (the product of Cd and frontal area of the vehicle) from on-road tests is presented. The possibility to measure these properties during an on-road test, without the need for a test track, enables third parties to verify the certified vehicle properties in order to preselect vehicle for further regulatory testing. On-road tests were performed with three heavy-duty vehicles, two lorries, and a coach, over different routes. Vehicles were instrumented with wheel torque sensors, wheel speed sensors, a GPS device, and a fuel flow sensor. *Cd·A* of each vehicle is determined from the test data with the proposed methodology and validated against their certified value. The methodology presents satisfactory repeatability with the error ranging from −21 to 5% and averaging approximately −6.8%. A sensitivity analysis demonstrates the possibility of using the tire energy efficiency label instead of the measured *RRC* to determine the air drag coefficient. Finally, on-road tests were simulated in the Vehicle Energy Consumption Calculation Tool with the obtained parameters, and the average difference in fuel consumption was found to be 2%.

**Keywords:** greenhouse gas emissions; $CO_2$ emissions; fuel consumption; road loads; resistance forces; air drag coefficient; rolling resistance coefficient; vehicle simulation





## 1. Introduction

Heavy-duty vehicles (HDV) remain the most heavily utilized mode for moving goods both in the European Union (EU), where it comprises 75% of the total inland freight transport [1], and the United States (US), where the corresponding share is close to 63% [2]. Consequently, HDV share in transport Carbon Dioxide ($CO_2$) emissions is high, comprising 5.6% of the total transport sector as reported by European Automobile Manufacturers Association [3]. This value is estimated to be 25% of overall transport greenhouse gas emissions (26.5% in the on-road share), while future projections show that it will comprise 32% of the on-road $CO_2$ emissions in 2030 [4]. To obtain a reduction of the $CO_2$ emissions from transport, policy measures for HDV were introduced recently in both regions [5,6].

Simulation software to calculate the vehicle's energy consumption and associated fuel consumption and $CO_2$ emissions are being used in both the EU (VECTO—Vehicle Energy Consumption Calculation Tool) and the US (GEM—Greenhouse Gas Emissions Model). These simulation tools require several inputs, including the product of the aerodynamic drag coefficient and the frontal area of the vehicle (*Cd·A*) as well as the tire rolling resistance

coefficient (*RRC*). These 2 vehicle properties together with the vehicle's inertia mass are essential to calculate the energy demand of the vehicle during real-world driving conditions since, together with the losses of the different drivetrain components, they determine the total energy that needs to be provided by the engine.

The importance of air-drag and rolling resistance is also high because they play a significant role in energy efficiency and their $CO_2$ footprint. Manufacturers put serious effort to improve the vehicle and tire characteristics. To quantify this, an analysis in the energy audit over representative test cycles showed that the aerodynamics estimated to account for the 15% and 20% of the total energy losses for the EU and the US, respectively [7]. The difference between the two regions is attributed to the higher vehicle's speeds in the US (the power demand due to the air-drag increases as the cube of velocity [8]). Furthermore, the results of a questionnaire distributed to various stakeholders in both EU and US, estimated a reduction up to 7–26.3% and 6–13% in the fuel consumption for a variety of technical improvements in the vehicle shape and in the tire technologies [9]. In other words, *Cd·A* and *RRC* are essential for benchmarking the energy efficiency among different manufacturers in the same vehicle category, and for quantifying the energy savings of new technologies, which will also support the engagement for reducing the greenhouse gas emissions in the transport sector. Moreover, combined with data about the vehicle share in the fleet and the driving patterns (vehicle average speeds and kilometers driven), the impact of HDVs on the environment can be analyzed.

Although the theoretical principles behind the derivation of beforementioned coefficients are well determined, it is difficult to measure them accurately. The tests are sensitive to a number of variables (e.g., environmental), while the test procedure requires high allocation of resources and can become relatively complex. Whereas a constant speed test is applied in the EU, a coast down test is applied in the US. It has been reported that among the two different procedures suggested by the US and EU there can be discrepancies in the results, reaching 12% in the drag coefficient calculation [10].

In the present study, a methodology to derive the *Cd·A* is proposed based on on-road test data measured with torque meters. The method relies on filtering and keeping the most solid parts of the trips that were initially performed for direct energy efficiency measurement. Although the proposed methodology can also be applied to determine the *RRC*, the available test data for validation did not include the required low-speed sections to accurately determine the *RRC*. The methodology is applied to three HDVs with different body shapes, and a sensitivity analysis is presented when the tire declared energy efficiency class of the tire is used.

## 2. Methodology

### 2.1. Background

Air drag and tire rolling resistance are 2 of several parameters that oppose the vehicle's forward movement. They are included in the road loads, the resistance forces that a vehicle experiences on the road and they are proportional to the vehicle's speed, as described by the second-order polynomial function in Equation (1) [11]. Several ways exist to derive the road loads of a vehicle. Measured forces corresponding to specific vehicle speeds, in a range that covers the minimum and maximum velocity of the vehicle, are fed to the polynomial. *F0*, *F1*, *F2* (also referred as road load coefficients) are obtained by minimizing the difference for all the speed, force pairs.

$$Fr(t) = F0 + F1 \cdot v(t) + F2 \cdot v(t)^2 \tag{1}$$

where:

| | |
|---|---|
| *Fr* | Deceleration Force (N) |
| *F0* | constant resistance (N) |
| *F1* | resistance proportional to vehicle speed (N/(km/h)) |
| *F2* | resistance proportional to speed squared (N/(km/h)$^2$) |
| *v* | Vehicle speed (km/h) |

From a theoretical point of view, the resistance forces are either constant and correspond to the tire rolling resistance, $F0R_{RR}$, or proportional to the second power of the velocity and correspond to the air drag resistance force, $F2R_d$. Equations (2) and (3) show the theoretical calculation of these 2 forces [12]. However, studies have pointed out that the tire rolling resistance can slightly increase as the velocity increases [13–15]. In addition, depending on the test methodology, drivetrain losses can also be measured, which are proportional to the velocity.

$$F0R_{RR} = RRC{\cdot}W \text{ (N)} \tag{2}$$

$$F2R_d(\text{t}) = 1/2{\cdot}\rho{\cdot}C_d{\cdot}A{\cdot}\left(\frac{v(t)}{3.6}\right)^2 \text{(N)} \tag{3}$$

where:

| | |
|---|---|
| $\rho$ | air density (kg/m$^3$) at reference conditions |
| $C_d$ | aerodynamic drag coefficient (-) |
| $A$ | frontal area of vehicle (m$^2$) |
| $v$ | vehicle speed (km/h) |
| $RRC$ | rolling resistance coefficient (-) |
| $W$ | weight of vehicle (N) |

A method that does not account for the drivetrain losses includes individual testing of each component that opposes the movement of the vehicle. The tire rolling resistance is measured on a steel test drum with predetermined vertical loads, speeds, and inflation pressures. Furthermore, a single condition is proposed for the need to produce data in a large array of tires (Simplified Standard Reference Condition [16]). The aerodynamic drag is measured in a wind tunnel. Both tests are held inside facilities where it is possible to control all parameters that could affect the results, such as environmental conditions (ambient temperature, wind speed, etc.). This procedure allows for high repeatability of the results and aids in parametric evaluation [17]. The disadvantage of this method is that the cost of these tests can be high, and they might not represent real driving conditions. A comparison between indoor and on-road measurement of *RRC* showed that the indoor derived values were significantly lower [18]. Except for measuring errors in road testing, the divergence was mainly attributed to the tire alignment (camber, toe) on the road, to uncontrolled environmental factors, and to the roughness of the actual asphalt of the road, which resulted in additional friction losses. The purpose of indoor testing is to give the influence of the individual component with high precision. For example, tire manufacturers are performing drum tests for their products since this is the mandatory procedure to determine the energy efficiency class of the tire.

Another test methodology is the coast-down method. It is frequently used in the $CO_2$ type approval procedure of light-duty vehicles (LDV) in Europe [11,19]. During the test, the vehicle is left to decelerate from a specified high speed. The times to decelerate in specific speed intervals are transformed to forces per velocity clusters. Three coefficients are derived by a least squared regression curve fitted to the data points (average velocity and average force pairs). The test is performed at a proving ground, where the impact of other parameters, such as traffic, is limited and the test path is well defined. The advantages include the need for less instrumentation, while it can achieve repeatable results [20]. On the other hand, this kind of tests is time-consuming and includes the drivetrain losses. Moreover, HDVs require a long distance to obtain a speed reduction. This test procedure does not give the possibility to derive the forces that are directly related to the tire deformation and the shape of the vehicle. Nevertheless, it is suitable for the LDV $CO_2$ regulation for deriving the road load coefficients for the emission tests in the laboratory [11]. A similar approach is used in the US HDV $CO_2$ certification methodology for deriving the tire rolling resistance and air drag coefficients. The coast-down is performed only for a high and a low-speed segment, allowing the use of short test tracks with good quality road surface. This can be highly important for getting comparable test results among different proving grounds. Additionally, a correction is applied to single out the speed-

dependence of the axle spin losses and of the tire rolling resistance [6]. In a study by the United States Environmental Protection Agency (US EPA, Washington, DC, USA) the methodology showed the ability to determine aerodynamic differences between tractors, and the effectiveness of devices to improve the aerodynamics. The standard error was below 2%, and the importance of at least 14 repetitions to reduce the uncertainty was pointed out [21]. In a study contacted by National Research Council Canada, where they tried to reproduce the *Cd·A* values calculated by the US EPA for a vehicle under the same configuration, the difference was found to be within 5% [22].

Other methodologies include wheel torque measurements. The method applied for the derivation of the *RRC* and *Cd·A* of HDVs in Europe uses wheel torque meters during constant speed tests at 2 steady velocities. One at very low speed (10–15 km/h), where the impact of the aerodynamic forces is considered negligible and the measured force is attributed to the tire rolling resistance. The second speed (85–95 km/h) is close to the maximum speed the vehicle can achieve, and the extra force measured is attributed to the aerodynamic resistance [5,23]. The advantages of this method include the exclusion of the drivetrain losses and its high repeatability [23,24]. Regarding repeatability, in [24], US EPA performed, in the same test site and with the same HDVs, both coast-down and constant speed test methodologies. The latter had lower standard deviation of the mean (under 0.5%), and lower dependency on the wind yaw angle. The downside is that it can be costly (the torque meter equipment) and requires a test track. Another approach that uses torque meters suggests that there is no need for steady speed tests, but the entire test's wheel force measurement can be fitted to a regressor and a second-order polynomial function. This investigation showed that the coefficients have a better agreement with the tractive force than the coefficients derived from the coast-down test [17]. More specifically, the coefficient of determination ($R^2$) was calculated to be 0.9926 and 0.9280 for the force method and the coast down, respectively.

The above-mentioned methodologies are well defined and provide adequate repeatability and reproducibility; however, they require specific equipment, instrumentation, and facilities. They are time-consuming and the test schedule cannot be combined with other activities. In a previous study [25], the authors investigated the derivation of the resistance forces of LDVs using torque meter data from on-road tests, under real-world conditions on public roads. Taking into account that environmental parameters—which can dramatically affect the results—were not considered at the study, the accuracy of ±3–7% for the aerodynamic resistance as well as the good agreement in fuel consumption pose a good base for the present study. One drawback of the methodology is the need for the type approval road loads as boundaries to filter out outliers. Unfortunately, these values are not available for HDVs.

### 2.2. Experimental Equipment

Three HDVs were tested on the road for the purposes of the present study. The tested vehicles belong to different HDV categories: Vehicle 1 is a heavy lorry (tractor with semi-trailer), vehicle 2 is a heavy bus (coach), and vehicle 3 is a medium lorry (rigid truck). Figure 1 shows an illustration of the vehicles and their technical specifications are listed in Table 1.

All vehicles were tested in standard operating condition without modifications, except for the addition of the necessary instrumentation. The instrumentation was powered by external batteries and was not connected to the vehicle. An overview of the instrumentation is provided in Table 2. The wheel torque and speed measurements were required to determine the vehicles' road loads, whereas the fuel consumption and engaged gear signal were measured additionally to simulate the trips in VECTO for the validation of the methodology. The accuracy of the measurement devices is critical to precisely determine the *Cd·A* coefficient and *RRC* with the proposed methodology. Test equipment uncertainty is not extensively discussed in this paper; previous research has run similar analyses [26–28] while EU regulation introduces necessary provisions. The torque measurement sensors

were zeroed by lifting the driven axle before the start of each test. The zeroing procedure was repeated at the end of the test, as foreseen by the official test procedure, to measure the drift of the torque sensors and a linear correction was applied to the measured torque signal to cancel out the error introduced by the drift.

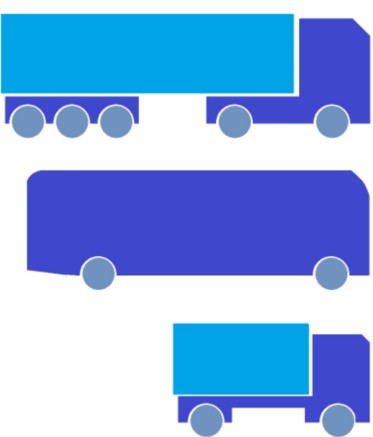

**Figure 1.** Shapes of the tested vehicles. **Top**: Vehicle 1. **Middle**: Vehicle 2. **Bottom**: Vehicle 3.

**Table 1.** Technical specifications of the vehicles tested.

|  | Vehicle 1 | Vehicle 2 | Vehicle 3 |
|---|---|---|---|
| Vehicle category (-) | N3 (heavy lorry) | M3 (heavy bus) | N2 (medium lorry) |
| Maximum mass (t) | 40.0 | 19.5 | 5.5 |
| Tested mass (t) | 33.6 | 16.8 | 4.6 |
| Engine displacement (l) | 12.8 | 10.7 | 3.0 |
| Rated power (kW) | 330 | 315 | 110 |
| Gearbox type (-) | AMT | AMT | MT |
| Emissions category (-) | EURO VI | EURO VI | EURO VI |

**Table 2.** Vehicle instrumentation.

|  | Vehicle 1 | Vehicle 2 | Vehicle 3 |
|---|---|---|---|
| Fuel consumption | Fuel flow meter | Fuel flow meter | Fuel flow meter |
| Wheel torque | Wheel rim sensor | Wheel rim sensor | Half shaft sensor |
| Wheel torque accuracy | 0.1% full scale | 0.1% full scale | 0.378% |
| Wheel speed | CAN | CAN | CAN |
| Engaged gear | CAN | CAN | CAN |
| Engine torque | CAN | CAN | CAN |
| Engine speed | CAN | CAN | CAN |
| Sampling rate (Hz) | 10 | 2 | 1 |

### 2.3. Test Cycles

Each vehicle was tested over a different route—or different routes—and each test was repeated multiple times. All routes contain motorway, rural, and urban driving and represent a typical use cases for each vehicle. Routes 2 and 4 have a higher share of highway driving, while Routes 3 and 5 have a higher share of urban and rural driving. The distance-based driving shares of each route are listed in Table 3 and were derived for each route from the mission profiles in VECTO, based on the vehicle category. The standard error (SE) of the positive wheel work, fuel consumption, and vehicle-specific fuel consumption (VSFC) indicate good repeatability of the tests. The VSFC is defined as the ratio of the fuel consumption to the positive wheel work.

**Table 3.** Route characteristics.

|  | Route 1 | Route 2 | Route 3 | Route 4 | Route 5 |
|---|---|---|---|---|---|
| Distance (km) | 108 | 153 | 103 | 153 | 68 |
| Urban distance share (%) | 17 | 9 | 30 | 9 | 59 |
| Rural distance share (%) | 28 | 10 | 35 | 15 | 14 |
| Motorway distance share (%) | 55 | 81 | 35 | 76 | 27 |
| Median speed (km/h) | 59 | 88 | 47 | 87 | 44 |
| SE wheel work (%) | 6.1 | 1.5 | 2.0 | 13.9 | 6.2 |
| SE fuel consumption (%) | 4.2 | 2.0 | 2.7 | 13.3 | 4.5 |
| SE VSFC (%) | 1.9 | 0.5 | 0.8 | 2.7 | 3.8 |
| Mean Ambient temperature (°C) | 10 | 19 | 16 | 10 | 11 |
| Number of Repetitions | 3 | 2 | 3 | 4 | 4 |
| Vehicle | 1 | 2 | 2 | 3 | 3 |

During the test, the auxiliary power demand was set to a minimum by switching off the air conditioning system and setting the ventilation to a minimum. Parts of the tests where the vehicles were not sufficiently warmed-up have been excluded from the analysis. For some vehicles, the effectiveness of the warm-up phase was confirmed by a measurement of the axle temperature. Moreover, there was no regeneration of the particle filter during the tests.

*2.4. Data Processing and Road Load Calculation*

The present study is based on the methodology described in [25] with several enhancements to make it more robust and more broadly used. The current approach does not apply boundaries to determine if a measured force in a specific vehicle speed is valid. These boundaries were set for LDV's, where the Original Equipment Manufacturer (*OEM*) official road load coefficients ($F0_{OEM}$, $F1_{OEM}$, $F2_{OEM}$) were available and experimentally derived. The methodology was applied to each trip individually.

The first step of the method includes the identification of the trip segments where the velocity can be considered constant. For that reason, the segments where the speed in each time step does not exceed the average velocity of the segment by more than 1 km/h are considered. This limit is justified by the fact that the speed measurements often come either from GPS, or On-board diagnostics (OBD) logging, so the accuracy can be limited. The derivation of the acceleration values from the low-resolution velocity signal can lead to dropping unrealistic values in segments that could be considered as constant. The selection of velocity rather than the acceleration signal as the mean for identifying the constant speed segments is also justified by the fact that velocity values are easier to interpret compared to acceleration ones.

With several segments of constant speed, the *steady speed segments*, the following sequence of actions is applied:

- Smoothing filter on the elevation recording (Savitzky Golay filter [29,30]) to improve the road grade calculation. This filter is applied in a window of 31 subsequent values. The window length value must be a positive odd integer [31], and is a compromise between the need for smoothing, and not distorting the original elevation shape. Applying a filter is important in cases of the low resolution of the GPS elevation recordings. After the filtering the elevation signal, but also the produced road grade, feature a more realistic pattern. Road grade values above 10% are considered invalid and an average of the last 4 recordings is assigned to them.
- The motive power is calculated along with the motive power without the impact of the road grade. The latter is the wheel power needed for the vehicle to drive the same velocity profile on a flat road.

The next steps are related to the procedure for the calculation of the road load coefficients. Low-quality data or errors in the recording, especially in the GPS elevation signal, can affect the results negatively. For this reason, a distance-based rolling mean

is applied creating multiple observations from each constant speed segment, the *steady speed rolling segments*. This means that from each *steady speed segment* several overlapping segments are created. In Figure 2, a simple schematic example is presented. This way the outliers that are defined as the average values of wheel power that fall outside one standard deviation from the median of the segment can be excluded. Distance-based rolling mean was selected rather than time or sample-based, primarily because of the origin of the outliers we want to exclude, which are related to the vehicle position (GPS signal), and also because a constant value can be used in the same manner for different steady segments of different speeds of a trip. For further filtering of outliers, the produced rolling segments are processed. Segments with elevation and power signals appear noisy are dropped. In detail, rolling segments are excluded when the standard deviation of the elevation signal or the wheel power is higher than 0.4 m and 15 kW, respectively. These 2 values were derived empirically during the data analysis.

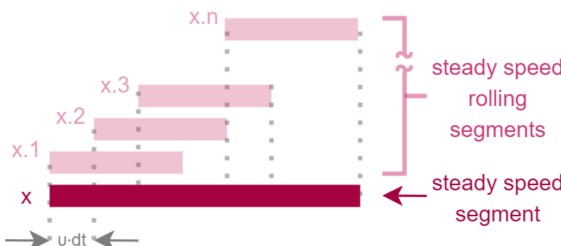

**Figure 2.** Schematic example of a steady speed segment, and the respective steady speed rolling segments. For each steady speed segment (x), various rolling segments (x.1, x.2, ... , x.n) are produced.

From the remaining rolling segments, the force per final *steady speed segment i* is calculated as:

$$F^i_{steady\ speed\ segment} = P^i_{steady\ speed\ segment} \Big/ \overline{U^i_{steady\ speed\ segment}} \tag{4}$$

where the $P^i_{steady\ speed\ segment}$ is the median wheel power value, without the impact of the road inclination, from all the rolling segments of each individual *steady speed segment*, *i*. $\overline{U_{steady\ speed\ segment}}$ is the average velocity of the segment *i* in m/s. The reason for using the average segment speed instead of the median is that from the above procedure the speed values are inside a very limited range, thus there are no outliers that could drive the value used.

Until this point, there is filtering inside each *steady speed segment*, but there is no interaction between the different ones at different speeds. It has been observed that the average force value of a *steady speed segment* can be unrealistic compared to the average value of a *steady speed segment* in a lower or higher speed. For this reason, extra filtering of outliers is applied, in which the average power values in the whole velocity range of the trip are grouped in velocity clusters. For that reason the K-Means clustering method is applied [32]. From each group, the values that fall outside one standard deviation from the median are removed. A schematic simple example is presented in Figure 3.

From the above steps, we consider that the pairs of velocity and power are robust, and any impact from a possible low signal quality, intervention of traffic, and wind sudden changes has been filtered out.

Then the fitting process follows with the aim of deriving the raw road load coefficients. The data passed to the fitting contain the median values of *steady speed rolling segments* of each *steady speed segment*. The measured power is at the wheel, so there is no impact from drivetrain losses. For this reason, resistances related to the first power of velocity (*F1* coefficient from Equation (1)) are limited to the range of ±0.1, to account for any small influence of the rolling resistance and air drag related to the velocity. Additionally, weights are added to the fitting of the points to determine the uncertainty of the force values

(1-dimension sigma parameter as defined in [33]). According to the weights, priority is given to the most robust force values, in the case that 2 (velocity, force) pairs have close velocity value but not realistic difference in force. The weights are produced as the product of the standard deviations of force, velocity and altitude, and the average speed of each point. The procedure up to this point is summarized in Figure 4.

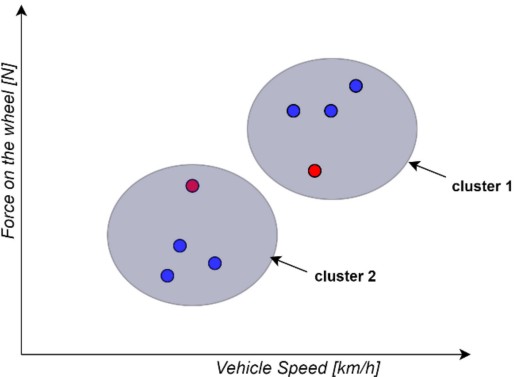

**Figure 3.** Schematic presentation of the filtering with K-means clustering. The dots correspond to the mean forces per steady speed segment. The red ones will be filtered out.

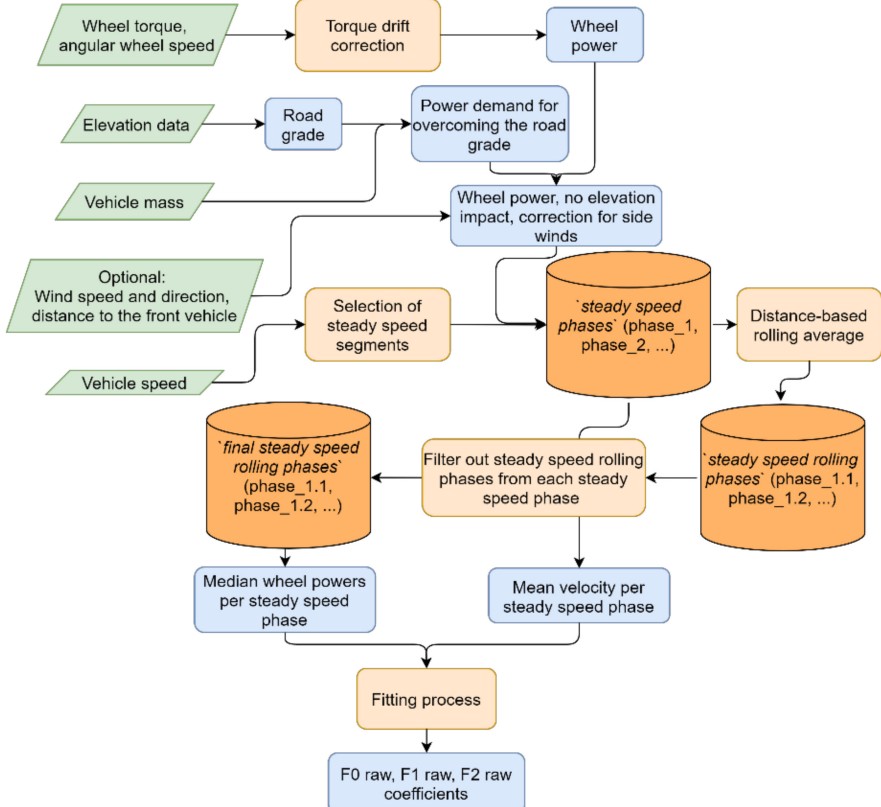

**Figure 4.** Data processing flow diagram.

As explained above, the distance-based approach is essential for filtering out the outliers and the last step is the selection of the most appropriate distance value for the *steady speed rolling phases*; appropriate in the sense of being long enough to contain segments with errors to be dropped. This distance value is selected after either an iterative procedure inside a range of values, or applying the minimization method Dual Annealing [34,35] to find the global minimum. The error function to minimize in both methods is the difference between the measured and simulated accumulated positive wheel power for the

parts of the test where speeds are below 35 km/h and above 80% of the maximum speed achieved during the test. In more detail, the simulated wheel power is calculated with the Equation (5):

$$P_{wheel}^{simulated}(t) = \left( m \cdot (1 + 4\%) \cdot \alpha(t) + F0_{raw} + F1_{raw} \cdot v(t) + F2_{raw} \cdot v^2(t) + m \cdot g \cdot sin\theta(t) \right) \cdot \frac{v}{3.6} \ (W) \tag{5}$$

where $F0_{raw}$, $F1_{raw}$, $F2_{raw}$ are the raw road load coefficients derived by the fitting process, m is the vehicle mass with an addition of 4% to account for the rotational mass. The extra 4% is derived by the default value proposed in the EU $CO_2$ type approval certification procedure of LDVs [11], with an additional 1% to account for the rotating mass of the wheel torque meters. The letter $\alpha$ represents the longitudinal vehicle acceleration, and $\theta$ is the road gradient. For the present study, the iterative procedure was selected, and the distance is obtained from a range of 40 to 140 m.

To derive the final test coefficients $F0_{test}$, $F1_{test}$, $F2_{test}$ and the *RRC* and *Cd·A* values to be compared with the Type Approved values, further corrections are needed to have the same reference. In detail, for the final Road Load test coefficients, the $F1_{test}$ coefficient is set 0 since in the present study we are dealing with HDVs. The procedure to recalculate the $F0_{test}$ and $F2_{test}$ is shown in Equations (6)–(11). It is a practice followed in the LDV EU $CO_2$ certification procedure in the vehicle road load derivation with the coast down method. It is used to exclude the influence of the *F1* factor when comparing 2 different vehicle configurations. This way, the difference in F0 can be attributed to the different tire efficiency, and the difference in the *F2* to the different *Cd·A*. The coast down times are collected with a step of 10 km/h starting from maximum speed of 130 km/h. For HDV vehicle, as max value we use the more realistic 110 km/h.

$$Speed_{referconce} = [10|110 \ km/h, step\ of\ 10\ km/h] \tag{6}$$

$$Force_{raw} = F0_{raw} + F1_{raw} \cdot Speed_{reference} + F2_{raw} \cdot Speed_{reference}^2 \tag{7}$$

Linear fitting of the $Force_{raw}$ to the speed squared values is applied, as presented in Equation (8):

$$Force_{raw} = \alpha + \beta \cdot Speed_{reference}^2 \tag{8}$$

The intercept ($\alpha$) and the slope ($\beta$) of the fitting are assigned to the *F0* and *F2*, respectively, ending to the results as:

$$F0_{test} = \alpha \tag{9}$$

$$F1_{test} = 0 \tag{10}$$

$$F2_{test} = \beta \tag{11}$$

These coefficients are not corrected for the environmental conditions during the test (e.g., ambient temperature correction). In principle, this means that the test derived road load coefficients can better represent the specific test in the specific environmental conditions and vehicle state. To use them in tests performed in the dyno, normalization of the $F0_{test}$, and $F2_{test}$ to the desired vehicle mass, and reference air density, respectively, should be applied.

The *RRC* can be considered independent of the vehicle speed, while the *Cd·A* is influenced only by the vehicle speed squared. We use the methodology described above in the Equations (6)–(11), with one difference: The range of speed references as described in Equation (6) contains only the first and last speed values: $Speed_{referconce} = [10,\ 110\ km/h]$. The reason for this choice is explained in detail in [25] and has to do with the influence of the original $F1_{raw}$ contribution. Using the full range of reference speeds as in Equation (6) for the linear fitting, the line produced by the adjusted road load coefficients ($F1_{test}$, $F1_{test}$, $F2_{test}$) might not pass through the experimentally derived points for low and high speeds, affecting the calculation of *RRC* and *Cd·A*, respectively.

The calculated $F0_{test}$ and $F2_{test}$ are set equal to the $F0R_{RR}$ and $F2R_d$, as defined in Equations (2) and (3), respectively. The air density used is the one reflecting the average test weather temperature. This way the coefficient $Cd{\cdot}A$ is not affected by weather conditions, and it can be directly compared to the *OEM*'s declared value.

## 2.5. Vehicle Simulation

VECTO is the vehicle simulation tool developed by the European Commission to certify the fuel consumption and $CO_2$ emissions of new heavy-duty vehicles [5]. In VECTO, the vehicle's longitudinal vehicle dynamics are simulated over a driving cycle based on the vehicle's technical properties (e.g., mass, air drag, tire rolling resistance) and a driver model. The instantaneous engine power is determined from the power demand at the wheels, the losses in each component of the powertrain and the auxiliary power demand. The engine speed is determined from the engaged gear, the powertrain gear ratios, and the dynamic tire radius. The engine torque and speed are used to interpolate from the engine fuel map to calculate the instantaneous fuel consumption. For the certification, the vehicle is simulated over pre-defined driving cycles, however for testing purposes, such as this investigation, custom driving cycles can be simulated with a target vehicle speed or wheel power.

The test cycles were simulated in VECTO with the calculated $Cd{\cdot}A$ and $RRC$ as input to validate the methodology. A model of each vehicle was built in the tool based on the available vehicle properties and with reverse-engineered efficiency maps for the axle and transmission and fuel consumption map for the engine. The measured vehicle speed, the engaged gear, and the road gradient derived from the GPS signal were given as input. All measurement signals were down- or up sampled to 2 Hz.

To validate the vehicle models, the test cycles were simulated in $P_{wheel}$ mode as well. In this case, the measured wheel power, the engine speed, and the engaged gear were given as input to VECTO. As a result, the driver model and vehicle road load parameters ($Cd{\cdot}A$, tire $RRC$ and vehicle mass) are bypassed, allowing a validation of the vehicle's powertrain. The error between the simulated and the measured fuel consumption in $P_{wheel}$ mode is less than 1.2% for each vehicle (Table 4), demonstrating an accurate model of the vehicle's power train.

**Table 4.** VECTO fuel consumption error using the $P_{wheel}$.

| Vehicle | Route | Error Simulated Fuel Consumption (%) |
|---------|-------|--------------------------------------|
| 1 | 1 | 0.75 |
| 2 | 2 | 0.33 |
| | 3 | −0.28 |
| 3 | 4 | −1.13 |
| | 5 | −0.32 |

## 2.6. Yaw Angle Correction

The European HDV $CO_2$ certification [5] proposes a zero yaw angle correction for the constant speed test results as a function the measured average yaw angle, $\beta$:

$$\Delta C_d A(t) = \alpha_1 {\cdot} \beta(t) + \alpha_2 {\cdot} \beta^2(t) + \alpha_3 {\cdot} \beta^3(t) \tag{12}$$

where the coefficients *α1*, *α2*, and *α3* are defined according to the vehicle segment, and they indicate the change of the vehicle area vertical to the wind speed vector. A schematic example of the vehicle effective frontal area that is considered according to the $\beta$ is shown in Figure 5. The cross winds during a test can have a significant impact for the HDVs where the effective area, depending on the yaw angle, can change dramatically, increasing the $Cd{\cdot}A$ product up to 3 m$^2$. For this reason, the same approach was adopted for vehicle 1

where the yaw angle was measured, correcting the instantaneous measured wheel power for the impact of the yaw angle according to Equation (13).

$$P^{measured}_{wheel,\ yaw\ angle\ corrected}(t) = P^{measured}_{wheel}(t) - \frac{1}{2} \cdot \rho \cdot \Delta C_d A(t) \cdot \left( \frac{v(t)}{3.6} \right)^3 \ [\text{W}] \qquad (13)$$

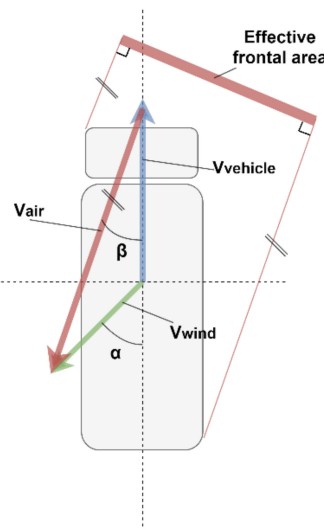

**Figure 5.** Schematic example of the effective frontal area to be considered for the aerodynamic drag coefficient (*Cd·A*).

## 3. Results and Discussion

Figure 6 demonstrates three examples of different types of tests applied for the analysis. In light green the overall velocity profile of the tests is presented. In dark green, together with the enumeration in red font, steady speed segments selected automatically are presented. Each segment fulfills the limit of maximum 1 km/h difference from the average speed of the segment. Figure 6 also depicts the recorded torque from the two wheels for the parts belonging to the steady speed segments.

The methodology proposed in the present study tries to identify both tire rolling resistance coefficient and air drag coefficient. However, the road tests were conducted in framework of the regulatory vehicle testing procedure for heavy duty vehicles [5]. Thus, due to the dynamicity of the test cycles, it was impossible to obtain observations in speeds lower than 35 km/h, where the impact of the air can be neglected. Lacking low-speed recordings renders the methodology unstable, lowering the accuracy in case sufficient data can be collected for intermediate speeds, or even not providing realistic values if the fitting must be performed using only high speeds segments. A work around for this case is fixing the $F0_{test}$ coefficient to values determined from the Equation (2) using as *RRC* default values.

The present methodology applies information from the tire label and values declared by the vehicle manufacturer to fix the lower point for the fitting process. For all vehicles, the base values are the ones the *OEM* declared and fall inside their respective energy efficiency class. Details for the tires are presented in Table 5.

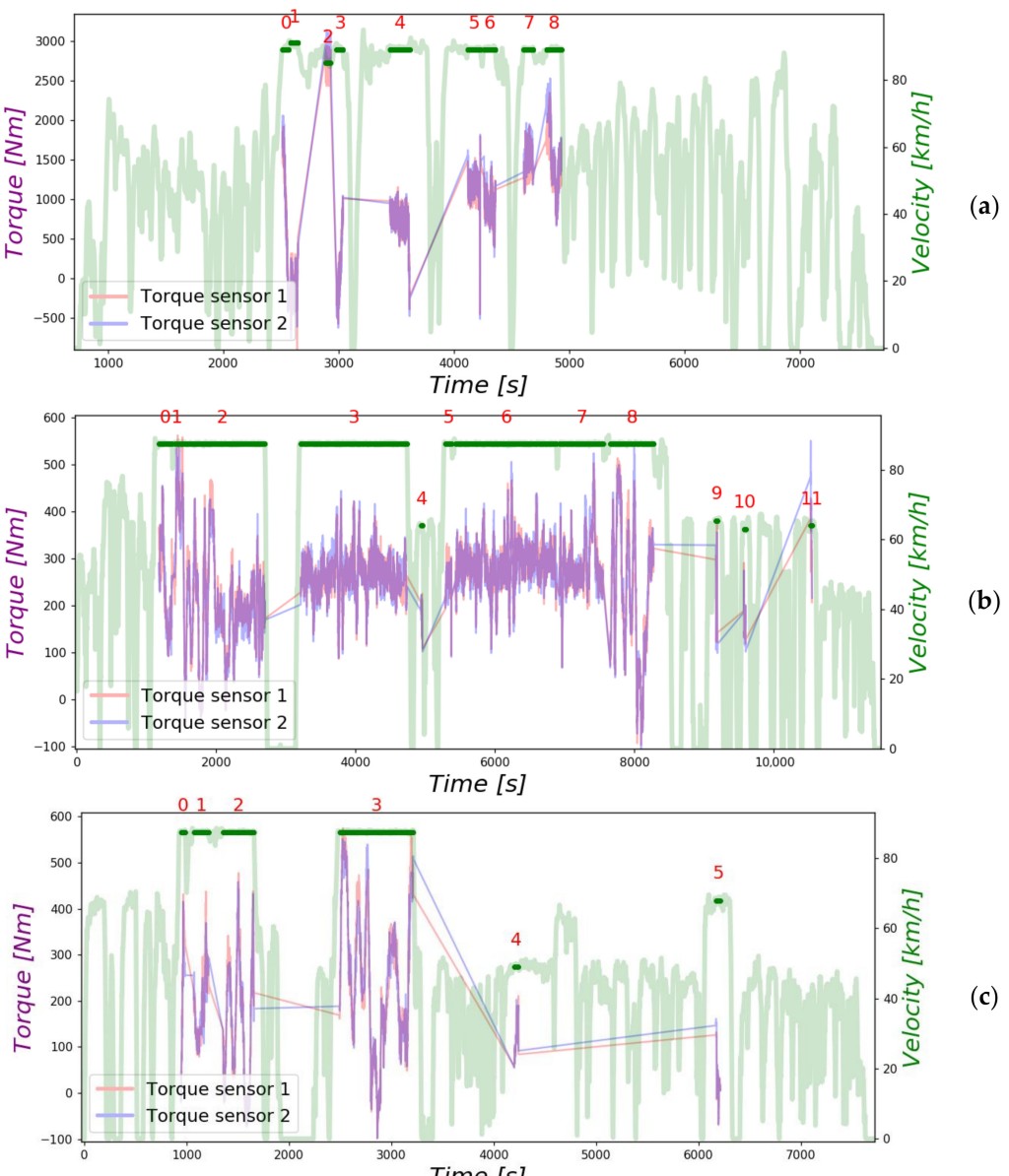

**Figure 6.** Examples of test types: (**a**) Combination of urban, rural, motorway; (**b**) motorway; (**c**) urban. The readings of torque sensor signals (light red and blue) refer to the left y-axis. The readings of the vehicle speed signal (light green), and the steady speed segments (dark green) refer to the right *y*-axis.

**Table 5.** Vehicle tires information.

| Vehicle | Vehicle 1 | Vehicle 2 | Vehicle 3 |
|---|---|---|---|
| Tire class | C3 | C2 | C3 |
| Energy efficiency class | C | B | C |
| Tire class range (kg/t) | 5.1–6.0 | 4.1–5.0 | 6.8–8.0 |

*3.1. Air-Drag Coefficient Calculation with Fixed Rolling Resistance Coefficient*

Figure 7 shows the error bars of the *Cd·A* value using the *RRC* values declared by the respective vehicle *OEM*s, as reported in Table 5. The error formula is defined as:

$$Error = \frac{CdA_{test}}{CdA_{OEM\ declared}} - 1 \tag{14}$$

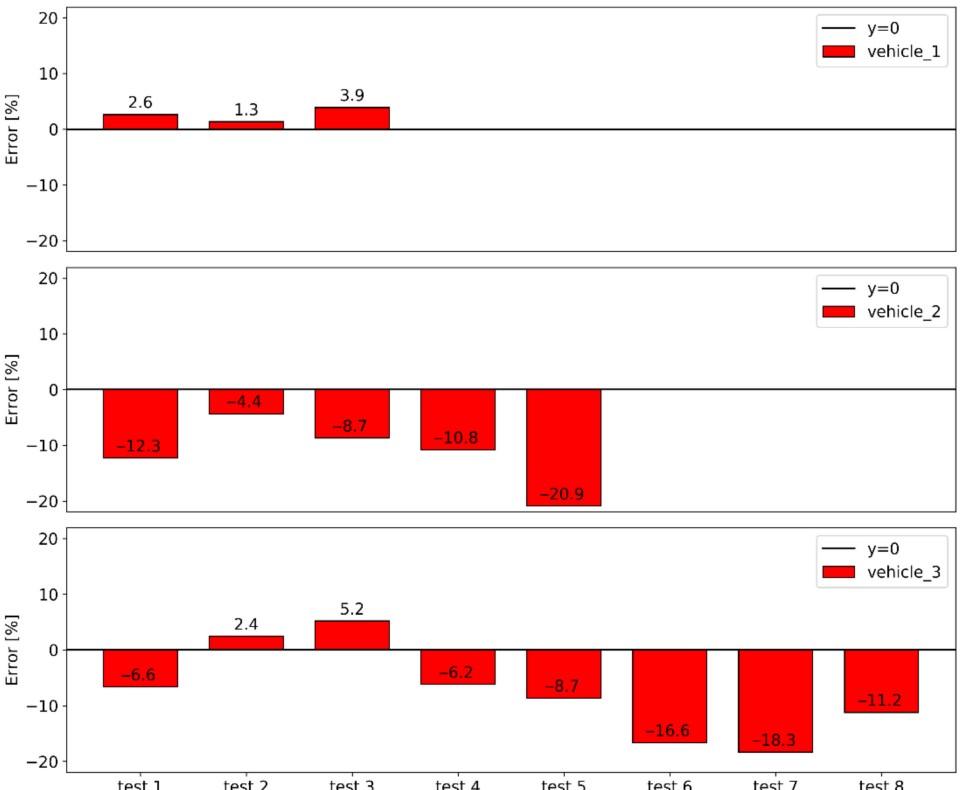

**Figure 7.** *Cd·A* Error bars per test.

For vehicle 1, the methodology over predicts the value that has been calculated in independent constant speed tests (according to the European HDV $CO_2$ certification). For vehicle 2 and 3, there is an underprediction compared to the values that the *OEM declares*, reaching approximately −21% and −18% for vehicles 2 and 3, respectively.

Table 6 shows the average and standard deviation of the error per vehicle. The best performance is achieved for vehicle 1 where the average and the standard deviation of the error are low (2.6% and 1.3%, respectively). For vehicle 2 the average error is −11.4%, the highest among the three vehicles, whereas the standard deviation of the error is found to be 6.1%. Vehicle 3 has an average error of −7.5% from the target *Cd·A* value and 8.3% standard deviation, the maximum among the tested vehicles. For a better validation of the method, the repeatability has been added to Table 6. As for repeatability we calculate the metrics relative standard deviation of the *Cd·A* values calculated by the methodology, and the standard deviation of the mean for each vehicle's error from the target value (percentage). Given the fact that at this point there are no further sources for correcting the uncertainty that the real world introduces, the standard deviation of the mean (0.8–2.9%) is considered satisfactory.

**Table 6.** Overall statistics per vehicle tested.

| Vehicle | Vehicle 1 | Vehicle 2 | Vehicle 3 |
|---|---|---|---|
| Average Error (%) | 2.6 | −11.4 | −7.5 |
| Error Standard deviation (%) | 1.3 | 6.1 | 8.3 |
| Error Standard deviation of the mean (%) | 0.75 | 2.72 | 2.92 |
| *Cd·A* relative Standard deviation (%) | 1.25 | 6.88 | 8.93 |
| Pooled standard deviation (%) | | 6.95 | |

The underestimation in vehicles 2 and 3 results in a total average error of −6.8%. The reason for the increased total average error might be that in public roads HDVs are obliged to limit their speed and have limited possibilities to overtake other vehicles. This can result

in a car-following scene that lowers power demand from the vehicles. The validity of this assumption will be evaluated through the fuel consumption comparison. Unfortunately, a distance-to-the-front signal was not available during the tests for vehicle 2 and 3. For vehicle 1, there were few cases were the distance to the front could impact the results, so there was no correction according to the distance-to-the-front performed. Taking into account all the factors that can impact the results, the standard deviation of the error for all the tests of approximately 8.2% is considered satisfactory. Considering the tests performed per vehicle as one set of tests, the pooled standard deviation is 7%. The distribution of the error is presented in Figure 8.

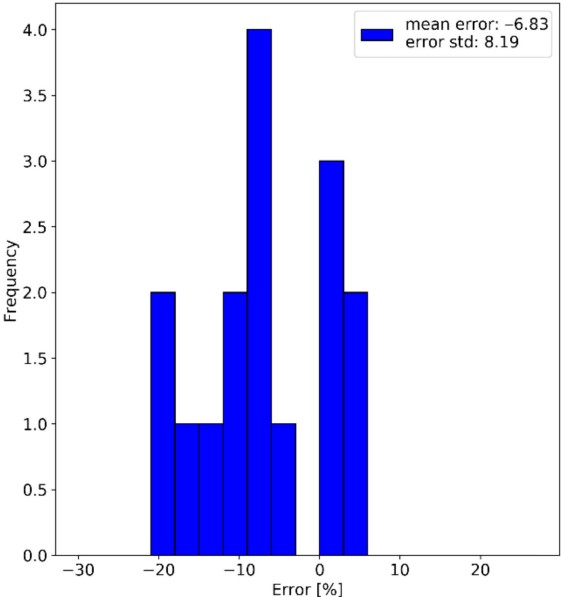

**Figure 8.** Overall *Cd·A* error distribution.

### 3.2. *Air-Drag Coefficient Calculation on Tire Label Rolling Resistance Coefficient Range*

The effect of the tire *RRC* on the obtained *Cd·A* was investigated with a sensitivity analysis. The minimum, average, and maximum values of the tire *RRC* within their energy efficiency class were used, as specified in Table 5 in accordance with the EU regulation [36]. The general observation is that as the *RRC* value increases the calculated *Cd·A* decreases. This is justified from the fact that using a higher *RRC* value we attribute a higher share to the tire resistance from the total value calculated, resulting in a lower impact of the air drag.

For vehicle 1 and 2, the impact of increasing the *RRC* by 0.5 kg/t results in a decrease of the average calculated *Cd·A* of approximately 5–6%. For vehicle 3 the reduction is limited to 2%. In all cases, the standard deviation is constant or affected slightly. A more descriptive presentation is placed in Figure 9, where the declared *Cd·A* values are compared to the ones produced by using the minimum, mean, and maximum energy efficiency of the tire class. Except for vehicle 2, that the average error for calculating the *Cd·A* is high in all different scenarios, from the results of vehicle 1 and vehicle 3, we can assume that even if we do not know the exact *RRC* value that the *OEM* declares for a specific vehicle, using the average value of the tire class will provide an acceptable *Cd·A* error (compared to the *OEM* declared).

Merging the results in Figure 10, we see that the lower average error for getting closer to the *OEM* declared *Cd·A* values is achieved by applying the lower value of the tire class. In detail the average errors are approximately −2.8%, −6.7%, and −10.7% for the minimum, average, and maximum values of the tire classes. The standard deviation for the minimum, average, and maximum values are similar, and above 9.5%, which is higher by more than 1% than by using the *OEM* declared *RRC*.

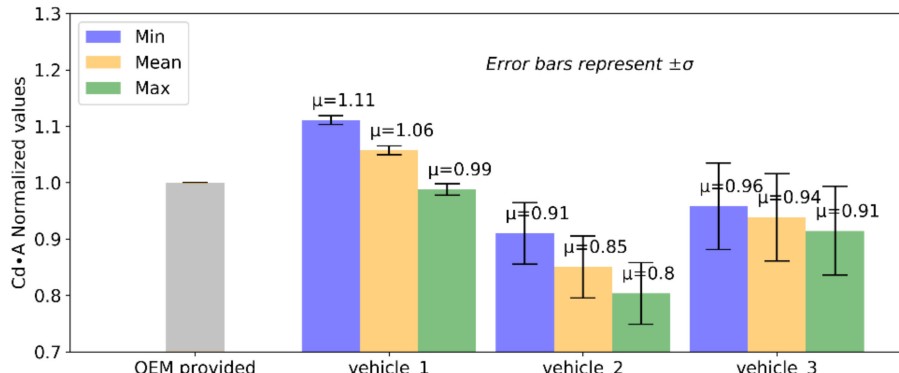

**Figure 9.** *Cd·A* calculated values for the range of tire class rolling resistance coefficient (*RRC*) ranges. The gray color bar represents the normalized *OEM Cd·A* value.

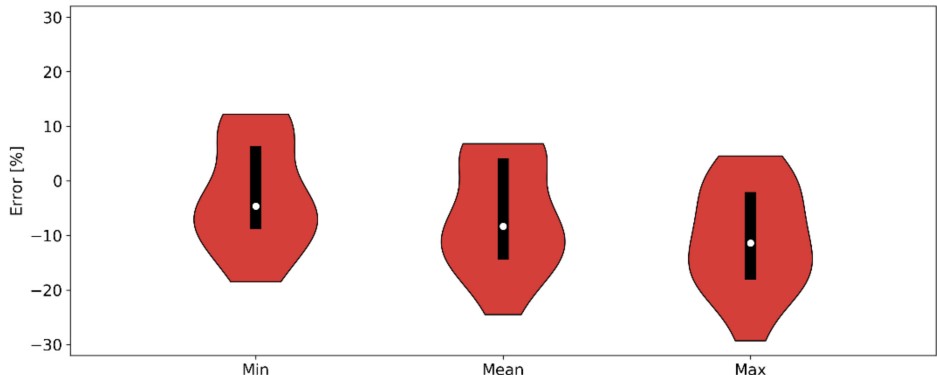

**Figure 10.** Violin plots for the total error distribution using the min, mean, and max *RRC* values of the tire efficiency classes. The white dots refer to the median values, while the thick black line corresponds to the range of the first to the third quartile.

### 3.3. Air-Drag Coefficient Calculation with Correction for Wind Yaw Angle

Wind recordings were available for the tests with Vehicle 1 and were used for assessing the wind correction regarding the wind yaw angle. The coefficients used for Equation (12) are presented in Table 7.

**Table 7.** Coefficients used for yaw angle correction.

| Coefficient | $\alpha1$ | $\alpha2$ | $\alpha3$ |
| --- | --- | --- | --- |
| Value | 0.03 | 0.0408 | $-0.0021$ |

The wind direction can have a non-negligible impact on the engine power required to overcome it. An example is shown in Figure 11, which takes into account the constant speed measurements with average speed of 91 km/h. In the left-hand side, the blue points represent the measured (uncorrected) wheel power values, while the red points show the values after the correction. Both parameters are plotted against the relative angle between the wind and vehicle velocity. The median values are 100 kW and 98 kW before and after the correction, respectively. The difference in the median values for the specific example is 2 kW, but as it is shown in the right-hand side of the same figure, depending on the yaw angle, the wind impact can be 12 kW for beta equal to 6 degrees. This means that the impact of side winds on the wheel power demand during this speed segment test could exceed 10%.

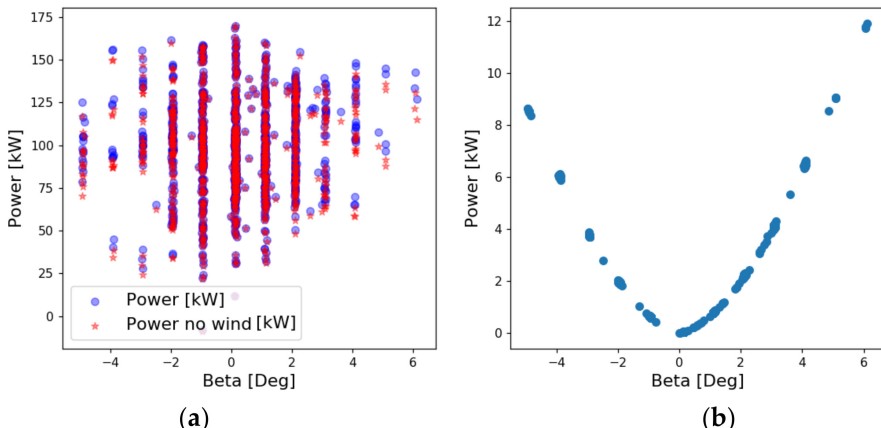

**Figure 11.** Wind correction example for a constant speed segment with 91 km/h average speed: (**a**) Power points before and after yaw correction; (**b**) power correction versus beta angle.

The correction for the three tests of vehicle 1 resulted in a mean error more centered around 0%, while the standard deviation slightly increased. This demonstrates the advantage of applying the yaw angle correction and measuring the yaw angle during the test. Details on the distribution and statistics are presented in Figure 12.

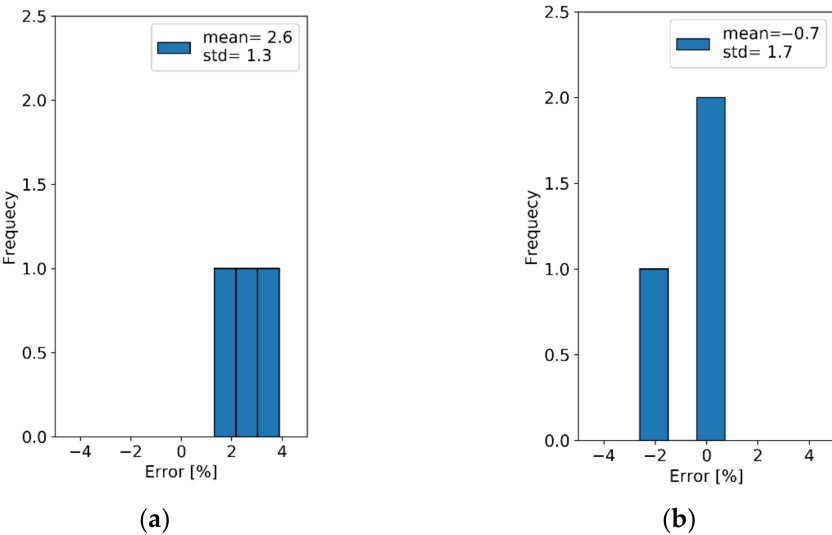

**Figure 12.** Overall results for vehicle 1: (**a**) Before correction; (**b**) after correction.

### 3.4. Fuel Consumption Simulation

The proposed methodology takes into consideration signals that may lack high accuracy (e.g., GPS altitude), and in its base case does not consider any interference from traffic or environmental conditions. The most demanding part of the methodology is the test data filtering to keep the most robust segments to derive the *Cd·A* of the vehicle. It also uses an iterative procedure to minimize the difference between the measured wheel power and the one produced by the road loads. However, the ultimate validation is the comparison of the measured fuel consumption versus the simulated, together with the positive wheel work of the whole trip. This is because the fuel flow meters can be considered as a highly accurate signal compared to the ones used in the methodology. Furthermore, it is not impacted by traffic conditions, while the impact of weather conditions is limited. The accurate VECTO vehicle simulation models, as presented in Table 4, supported this direction.

For each vehicle we use one average *Cd·A* value from the tests performed and we compare the results versus the declared values. From the set of *Cd·A* values used for the averaging for each vehicle we excluded the value calculated from the last test of vehicle

2, since it can be considered as an outlier. In Table 8, the percentage difference between the declared and the average values used per vehicle are given. In Figures 13 and 14, the error bars between measured and simulated fuel consumption and the wheel work are presented. For vehicle 1 each of the three tests performed in highway and urban parts is split into two subtests, resulting in a total of six tests. More details regarding the dominant share of each test for each vehicle is presented in Table 9.

**Table 8.** The ratio between mean calculated and declared values used for the VECTO simulations.

| | **Vehicle 1** | **Vehicle 2** | **Vehicle 3** |
|---|---|---|---|
| $\frac{Cd \cdot A_{mean\ calculated}}{Cd \cdot A_{declared}}$ | 1.0256 | 0.91 | 0.923 |

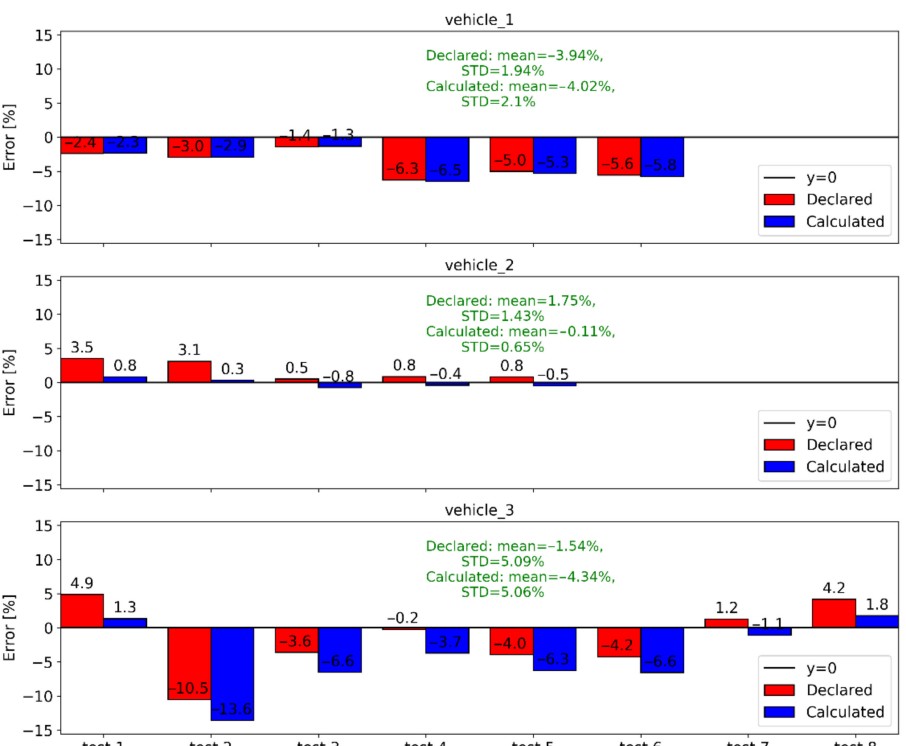

**Figure 13.** Fuel measured error using as input to the model the declared, and the calculated $Cd \cdot A$.

Starting from vehicle 1, simulations with both declared and calculated $Cd \cdot A$ are underpredicting the fuel consumption with a mean error of approximately −4%. A similar trend is shown for the wheel work error with both (declared and calculated) $Cd \cdot A$ values resulting in an average error of approximately −6.5%. Taking into account that the calculated value was 2.6% higher than the declared, the difference in $Cd \cdot A$ is low. The standard deviation is low for both signals, close to 2% for the fuel consumption, and 1.5% for the wheel work. The low standard deviation is following the low standard deviation achieved in the $Cd \cdot A$ calculation, proving the high precision of the data collected.

The calculated value of the $Cd \cdot A$ used for vehicle 2 is 9% lower than the declared and this difference is represented mostly in the first two tests where 80% of the total trip is driven on the motorway. This results in a difference of 2.7% in fuel consumption and 3.5% in wheel work. The average error in fuel measured is 1.8% for the declared $Cd \cdot A$, and −0.1% for the mean calculated. A similar picture is observed for wheel-work, where the error using the declared and calculated value is 1.6% and −0.7%, respectively. The standard deviation for the declared value is low. More specifically, it is below 1.5% for fuel measured and approximately 1.8 for the wheel work. Still, using the mean calculated $Cd \cdot A$ resulted

in a standard deviation of almost half compared to the declared. This demonstrates that the methodology accurately predicts the $Cd \cdot A$ of vehicle 2 during the test.

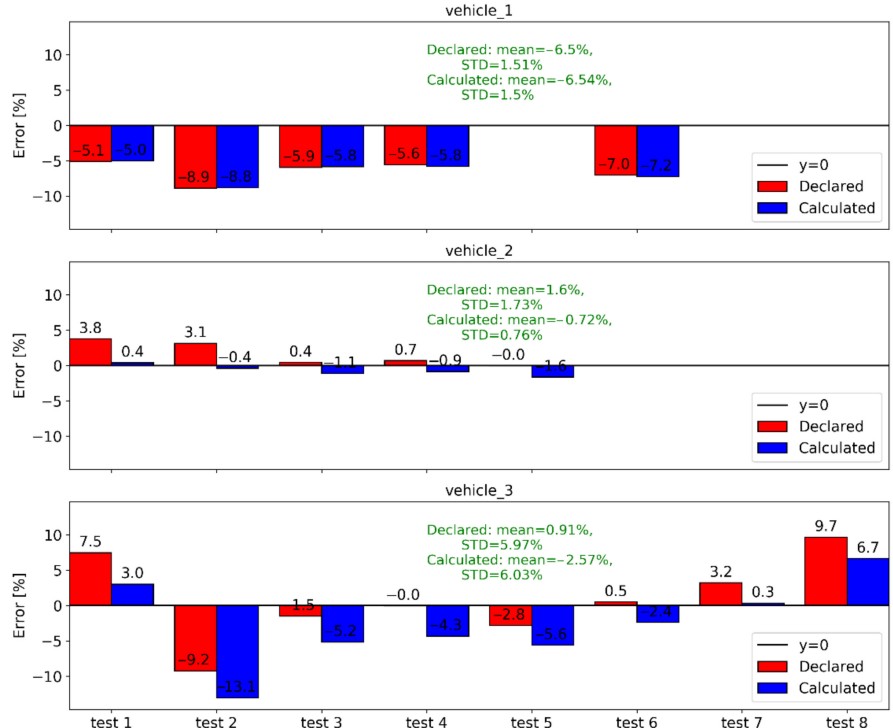

**Figure 14.** Wheel work error using as input to the model the declared, and the calculated $Cd \cdot A$.

**Table 9.** Test description according to the dominant share of the trip. M: Motorway, U: Urban.

| Test Number | Vehicle 1 | Vehicle 2 | Vehicle 3 |
|:---:|:---:|:---:|:---:|
| 1 | M | M | M |
| 2 | M | M | M |
| 3 | M | U | M |
| 4 | U | U | M |
| 5 | U | U | U |
| 6 | U | - | U |
| 7 | - | - | U |
| 8 | - | - | U |

Going to the third vehicle, where the mean calculated $Cd \cdot A$ value used is 7.7% lower than the declared, the differences are higher and reach even 4% in individual tests in the signal used for comparison. On average, the declared values underestimate the fuel consumption and wheel work measured by 1.5% and 1.7%, respectively, while the calculated value of $Cd \cdot A$ underestimates by 4.3% and 2.6% the fuel and the wheel work, respectively. Regarding the standard deviation of the two signals, both declared and calculated give similar values, close to 5% for the fuel and 6% for the wheel work. This demonstrates that the methodology accurately predicts the $Cd \cdot A$ of vehicle 3 during the test.

Overall results, all tests of all tested vehicles are presented in Figure 15. The mean error values for the fuel measured and the wheel work are −1.4% and −1.0%, respectively, when using the declared $Cd \cdot A$ of the vehicles. These errors are closer to zero errors compared to the calculated ones since the latter give a mean error of −3.1% and −3.2% for the fuel and the wheel work, respectively. This additional 2% of underprediction of fuel consumption and wheel work by using the calculated $Cd \cdot A$ is not corresponding to the almost 7% underprediction of the declared $Cd \cdot A$ values by the calculated ones. This can be explained by the fact that the calculated $Cd \cdot A$ values in some real-world tests can produce more

representative road loads than the declared ones, resulting in better agreement in the wheel work and fuel measured (e.g., vehicle 2). This can also be supported by the slightly lower standard deviation of the calculated $Cd·A$, which, in addition, is a good sign of the stability of the methodology. Another factor to account for is the share of high-speed segments, where the aerodynamics have a major contribution to fuel consumption. However, no connection was found in the presented study.

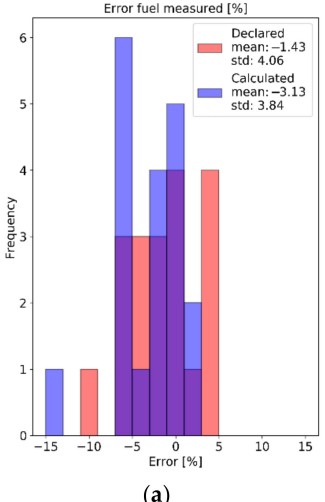
(**a**)

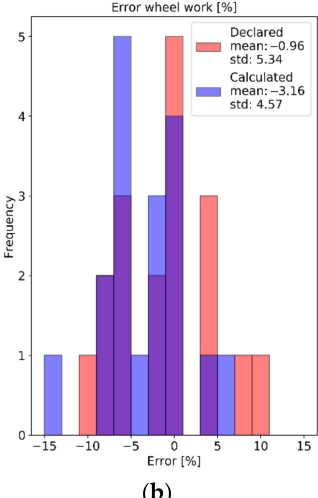
(**b**)

**Figure 15.** Overall error results for: (**a**) The fuel and (**b**) the wheel work for the declared and mean calculated $Cd·A$.

Finally, in Figure 16, we compare the wheel-work error obtained by VECTO simulations with the percentage difference between the average values used in the simulator and the $Cd·A$ values calculated by the individual tests. A test $Cd·A$ value lower than the average value used in the simulations would lead to an overprediction, while a higher test $Cd·A$ than the average would lead to an underprediction of the wheel work. Although this trend is visible in most of the values (the expected region of the dots is colored in light blue), the correlation is low for vehicles 2 and 3, as calculated with the use of $R^2$ metric. More details are provided in Table 10.

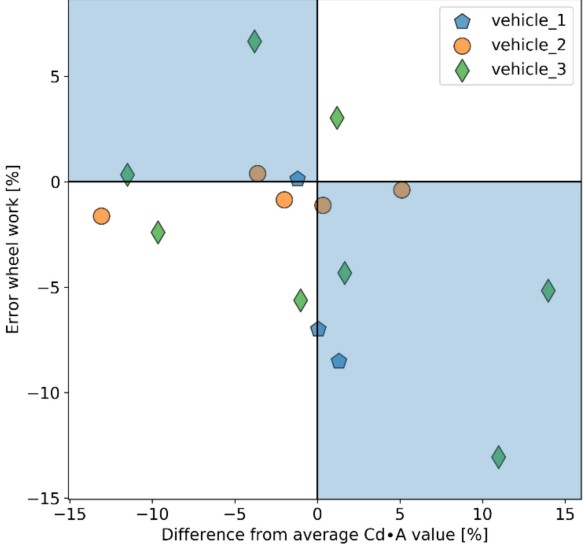

**Figure 16.** Plot to visually show the correlation between the error of the wheel work [%] and the percentage difference between the $Cd·A$ value calculated from the individual test and the average $Cd·A$ value.

**Table 10.** Linear regression coefficients fitted in the difference of the average $Cd{\cdot}A$ value used for VECTO simulations versus the error in wheel work of the tests.

| $y=a{\cdot}x+b$ | $\alpha$ | b | $R^2$ |
|---|---|---|---|
| Vehicle 1 | −3.44 | −5 | 0.87 |
| Vehicle 2 | 0.05 | −0.5 | 0.22 |
| Vehicle 3 | −0.38 | −2.5 | 0.33 |

## 4. Conclusions

Studies show the high contribution of *RRC* and $Cd{\cdot}A$ to the vehicle energy consumption, which is linked to the greenhouse gas emissions, and the importance to reduce their effect. The present work proposes a methodology for deriving the *RRC* and the $Cd{\cdot}A$ values of a vehicle under real-world operating conditions. The advantage of the method is that the test activities needed for the derivation of the two coefficients can run in parallel to other activities and possibly regular vehicle use. Hence, a possible use case of such on-road tests when combined with the proper equipment can be used to verify the officially declared $Cd{\cdot}A$ and tire *RRC* of a vehicle.

Results for the $Cd{\cdot}A$ values are presented from applying the method in three different HDV body categories, and they are compared to the ones declared by the *OEM*s. The average $Cd{\cdot}A$ error for the total number of 16 measurements is −6.8%, and regarding the repeatability, the pooled standard deviation is 7%. Due to the dynamicity of the tests, it was not possible to derive the *RRC* values, since there were not sufficient recordings in low speeds, so for the calculations, the official declared values were used. A sensitivity analysis on using the *RRC* from the tire class is also presented showing that the lowest values of the classes were giving an average error of −2.8% in $Cd{\cdot}A$. Finally, the use of the wind yaw angle in one of the vehicles led to the average error to drop by 2%.

In addition to comparing to the official values of $Cd{\cdot}A$, validation of the fuel consumption with the use of a simulation tool was performed. We compare the measured fuel consumption with the fuel consumption simulated using the declared $Cd{\cdot}A$ and the average test $Cd{\cdot}A$ per vehicle derived with the presented methodology. The fuel consumption and the positive wheel work were simulated with an error of 3.1 and 3.2%, respectively, with the calculated $Cd{\cdot}A$. The difference in fuel consumption measured between the declared and the average test $Cd{\cdot}A$ is 2%, and the error standard deviation is slightly smaller with the test values. In more detail, there are several simulated tests that had better agreement when the calculated $Cd{\cdot}A$ was used. This can be attributed to the fact that the test values can correspond better to the specific external factors that impact real-world driving, and the particularities of each test.

**Author Contributions:** Conceptualization, D.K. and G.F.; data curation, D.K., S.B. and T.G.; formal analysis, D.K. and S.B.; investigation, D.K.; methodology, D.K. and G.F.; project administration, G.F.; software, D.K., S.B. and T.G.; supervision, L.N. and G.F.; validation, D.K., S.B. and T.G.; visualization, D.K.; writing—original draft, D.K. and S.B.; writing—review and editing, D.K., S.B., L.N. and G.F. All authors have read and agreed to the published version of the manuscript.

**Funding:** This research received no external funding.

**Institutional Review Board Statement:** Approved for publication.

**Informed Consent Statement:** Not applicable.

**Data Availability Statement:** The data presented in this study are currently available on request from the corresponding author. They will become available in https://data.jrc.ec.europa.eu/.

**Acknowledgments:** The authors acknowledge the JRC Vehicle Emissions Laboratory (VELA) team for their support during the whole experimental activity described in the present study. Authors would like to thank Nikolaus Steininger and Hans Holdik for critically reviewing the manuscript.

**Conflicts of Interest:** The authors declare no conflict of interest.

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
