# Peer review of "In Use Determination of Aerodynamic and Rolling Resistances of Heavy-Duty Vehicles"

_sustainability, doi:10.3390/su13020974_

Round 1

Reviewer 1 Report

This paper proposes a new methodology for the measurement of the rolling resistance and the CdA coefficients of a heavy vehicle by on road test.

The method is based on torque and speed measurement at the vehicle wheel; the air drag and rolling resistance coefficients can be derived by means of proper regression techniques.

This implies that 1) the measuring instruments must be accurate; 2) that the measured values ​​must be properly filtered to eliminate the influence of disturbances.

While the authors show adequate statistical and filtration methods, as regards the first point the authors do not report the accuracies of the measuring instruments, nor do they perform the error analysis of CdA coefficient. Knowing the measurement error of CdA is important, especially when this coefficient must be compared with reference values.

The authors should clarify this point.

Finally, it is not clear why low speed tests were not carried out to calculate CCR as well.

typing errors

row 56: shown --> showed

Eq. 5: in the last term the speed is not squared

Row 410: figure 1 --> figure 10; the same in related caption

Fig.11 inside the plot Kw --> kW

The authors should have checked in advance the correct citation of the figures, so as to avoid annoying references to unidentifiable figures for the reviewer.

Author Response

We would like to thank both reviewers for the thorough review of the manuscript. It really helped us improve it and enhance it with additional material.

We have addressed all your comments in the manuscript, highlighted the changes, and replied to your questions in the rebuttal.

Kind regards,
the authors.

Reviewer 1

Comment: This paper proposes a new methodology for the measurement of the rolling resistance and the CdA coefficients of a heavy vehicle by on road test.

The method is based on torque and speed measurement at the vehicle wheel; the air drag and rolling resistance coefficients can be derived by means of proper regression techniques.

This implies that 1) the measuring instruments must be accurate; 2) that the measured values ​​must be properly filtered to eliminate the influence of disturbances.

While the authors show adequate statistical and filtration methods, as regards the first point the authors do not report the accuracies of the measuring instruments, nor do they perform the error analysis of CdA coefficient. Knowing the measurement error of CdA is important, especially when this coefficient must be compared with reference values. The authors should clarify this point.

Response: We agree that the accuracy of the measurement devices is important to precisely determine the CdA coefficient and RRC with the proposed methodology. The issue of the test equipment uncertainty is difficult to cover in this paper, as it would make the manuscript very extensive, and similar analyses have been done by other researchers. We add the text with respective references in the `Experimental equipment` chapter, now reading as follows:

“An overview of the instrumentation is provided in Table 2. The wheel torque and speed measurements were required to determine the vehicles’ road loads, whereas the fuel consumption and engaged gear signal were measured additionally to simulate the trips in VECTO for the validation of the methodology. The accuracy of the measurement devices is critical to precisely determine the Cd∙A coefficient and RRC with the proposed methodology. Test equipment uncertainty is not extensively discussed in this paper; previous research has run similar analyses [26–28], while EU regulation introduces necessary provisions.”

The measurement accuracy of the torque sensors has been added to Table 2. Unfortunately, the measurement accuracy of the wheel speed is not known to the authors, as the signal was obtained from the vehicle CAN. Therefore, no error propagation analysis was performed, and it was chosen to follow a more holistic approach by comparing the calculated CdA with the reference CdA in sections 3.1 – 3.3.

Comment: Finally, it is not clear why low speed tests were not carried out to calculate CCR as well.

Response: The road tests were conducted in framework of the regulatory vehicle testing procedure for heavy duty vehicles (EU 2019/318). The data analysis revealed that this test procedure does not contain steady speed segments at a sufficiently low speed to accurately determine the RRC with the proposed methodology. The above information is added in the manuscript, resulting to the text in the second paragraph in the `Results and Discussion` section:

“The methodology proposed in the present study tries to identify both tire rolling resistance coefficient and air drag coefficient. However, the road tests were conducted in framework of the regulatory vehicle testing procedure for heavy duty vehicles [5]. Thus, due to the dynamicity of the test cycles, it was impossible to obtain observations in speeds lower than 35 km/h, where the impact of the air can be neglected.”  

typing errors

row 56: shown --> showed

Eq. 5: in the last term the speed is not squared

Row 410: figure 1 --> figure 10; the same in related caption

Fig.11 inside the plot Kw --> kW

The authors should have checked in advance the correct citation of the figures, so as to avoid annoying references to unidentifiable figures for the reviewer.

Response: References and typing errors were corrected as proposed by the reviewer. We are sorry for any inconvenience caused by the broken links.

Reviewer 2 Report

Dear authors,

your paper is very interesting and deals with a very important topic.

In my opnion, your paper is well-written. Nevertheless, you need to correct the following points:

  • Line 22-23: "3" vs. "two" => please use uniform spelling ("3" and "2" or "three" and "two". Please check you whole paper (e.g. line 77).
  • In my opinion, the use of abbreviations in abstracts is not reader-friendly.
  • Line 53: You have already introduced the abbreviations for "air-drag and rolling resistance". Hence, use the abbreviations.
  • Line 55-58: Please highlight the correlation between vehicle speed and energy consumption => energy consumption = v^3 * A Cd * 1/2* ...
  • Line 76: "did" or "do"?
  • Euqation 1: Please insert a reference for this equation (e.g. line 82-88)
  • Line 119: What means "large distance"?
  • Line 131-136: Please insert a reference for this information.
  • Line 157 and the following parts of the paper: Please check the references for your figures and tables. I can not unterstand, that you have submitted a paper with these errors! In addition, on p. 8 there is a large empty space and the table 7 on p. 15 and 16 goes over two pages. Figure 12 is on p. 16, but the caption for this figure is on p.17. Therefore, please check you layout.
  • Line 155-158: Please introduce the abbreviation "vehicle 1", "vehicle 2", and "vehicle 3" in the text and not only below Fig. 2.
  • Table 2: Can you explain, how you choosed the different sampling rates? Does the sampling rate depends on the used hardware or what is the reason for the different sampling rates?
  • Line 175-...: The vehicles were tested on different routes. This is an good approach to get realisitic results. Nevertheless, how do you select these routes? Please explain the selection of the analyzed routes.
  • Line 206: Why do you applied the filter in a window of 31 subsequent values and not 30 or 32 subsequent values?
  • Line 225: Please check the blanks between "with" and "elevation".
  • Equation 4: Can you please introduce abbreviations for the used paramaters, e.g. Fi_steady speed segment => Fi_sss. This only an suggestions, but the names of the parameters are very long.
  • Line 258: Please check the blank between "in" and "[25]".
  • Figure 4: Please start with an upper letter in the green boxes, compare e.g. "evelation data" with "Vehicle mass".
  • Figure 4: The blue box "median wheel powers..." needs to be increased.
  • Figure 4: "Correction of the side winds" => does this mean, that the wind from the front is not relevant?
  • Line 275: What is the reason for the used correction value of 4%? Please explain, why you use 4%.
  • Equations 6-8: Please use abbreviation for the parameters.
  • Equation 6: "step of 10 km/h" => Why?
  • Line 286: "Road Load coefficients" vs. "road load coefficients". Please check the upper letters.
  • Line 293: It is similar to my comments above. You use some data or boundaries without an explanation. The missing explinations are the main reason for my assessment "Must be improved" regarding the "Are the methods adequately described".
  • Why 10 and 100 km/h?
  • Line 315: "2 Hz". Why you do not use a greater sampling rate? Is the sampling rate limited by the use hardware?
  • Line 321: You stated that your results are demonstrating an accurate vehicle model. Why is it accurate? Is your method / model accurate because it is more precise compared to existing methods?
  • Figure 6: Please make lines in legend thicker => "Torque sensor 1" and "Torque sensor 2"
  • Line 346-348: Please explain it in detail, why it is impossible to obtain observations in speeds lower than 35 km/h.
  • Line 354: Introduce the abbreviation "OEM".
  • Line 357: Please do not use abbreviations in subtitles.
  • Line 379-381: You say "might be" and "can". How would you like to identify the real reason for the average error of -6.8%? May be your method is not suitable for the shown vehicles as well as routes. 
  • Line 416: "Figure 1" => "Figure 10"
  • Line 425-426: Why 91 km/h and not 90 or 92 km/h etc.?
  • Figure 12: Increase the label "error [%]" of the x-axis.
  • Table 8: Can you use abbreviations for the paramerters?
  • Line 485-486: "4.3" and "%" are in two different lines.
  • Line 527: Please check the blank between "." and "The".
  • Line 627-634: Please do not use Wikipedia as a reference for the filter etc.

In addition, the inegration of more literature (e.g. 3-4 paper) regarding common methods for the determination of aerodynamic and rolling restistances would be improve your paper. Based on these references, you are able to highligt the weakness of these methods and the strenghthes of your method.

Author Response

We would like to thank both reviewers for the thorough review of the manuscript. It really helped us improve it and enhance it with additional material.

We have addressed all your comments in the manuscript, highlighted the changes, and replied to your questions in the rebuttal.

Kind regards,
the authors.

Reviewer 2

Dear authors,

your paper is very interesting and deals with a very important topic.

In my opinion, your paper is well-written. Nevertheless, you need to correct the following points:

  • Line 22-23: "3" vs. "two" => please use uniform spelling ("3" and "2" or "three" and "two". Please check you whole paper (e.g. line 77).

    Corrected in whole document. All words are turned into numbers.

  • In my opinion, the use of abbreviations in abstracts is not reader-friendly.

    We removed from the abstract the abbreviations of terms appearing only once. 

  • Line 53: You have already introduced the abbreviations for "air-drag and rolling resistance". Hence, use the abbreviations.

    We have introduced abbreviations for their respective coefficients. In the specific lines we refer to the forces, for which we have not added any abbreviation.

  • Line 55-58: Please highlight the correlation between vehicle speed and energy consumption => energy consumption = v^3 * A Cd * 1/2* ...

    We added in line 58 explanation according to your indications, and also a reference. The phrase is turned to:

    “The difference between the 2 regions is attributed to the higher vehicle’s speeds in the US (the power demand due to the air-drag increases as the cube of velocity [8]).”
  • Line 76: "did" or "do"?

    “Did” should be ok. For better understanding we changed the text a bit. The phrase is turned to: “Although the proposed methodology can also be applied to determine the RRC, the available test data for validation did not include the required low-speed sections to accurately determine the RRC.”

  • Euqation 1: Please insert a reference for this equation (e.g. line 82-88)

    Added reference in the text above. The phrase that includes it is turned to:

    “Air drag and tire rolling resistance are 2 of several parameters that oppose the vehicle’s forward movement. They are included in the road loads, the resistance forces that a vehicle experiences on the road and they are proportional to the vehicle’s speed, as described by the second order polynomial function in equation (1) [11].”

  • Line 119: What means "large distance"?

    We rephrased to “long distance”. The phrase is turned to:

    “On the other hand, this kind of tests is time-consuming and includes the drivetrain losses. Moreover, HDVs require a long distance to obtain a speed reduction.”

  • Line 131-136: Please insert a reference for this information.

    Added reference. The phrases in the text are turned to

    “One at very low speed (10-15 km/h), where the impact of the aerodynamic forces is considered negligible and the measured force is attributed to the tire rolling resistance. The second speed (85-95 km/h) is close to the maximum speed the vehicle can achieve, and the extra force measured is attributed to the aerodynamic resistance [5,23].”

  • Line 157 and the following parts of the paper: Please check the references for your figures and tables. I can not unterstand, that you have submitted a paper with these errors! In addition, on p. 8 there is a large empty space and the table 7 on p. 15 and 16 goes over two pages. Figure 12 is on p. 16, but the caption for this figure is on p.17. Therefore, please check you layout.

    Fixed. We are sorry for any inconvenience caused by the broken links.

  • Line 155-158: Please introduce the abbreviation "vehicle 1", "vehicle 2", and "vehicle 3" in the text and not only below Fig. 2.

    We have added the respective vehicle numbers in the description of the tested vehicles in the in the explanation of the equipment. The text there is turned to:

     “Three HDVs were tested on the road for the purposes of the present study. The tested vehicles belong to different HDV categories: vehicle 1 is a heavy lorry (tractor with semi-trailer), vehicle 2 is a heavy bus (coach), and vehicle 3 is a medium lorry (rigid truck)”.

    We take out the description of the vehicles in the legend of figure 2.

  • Table 2: Can you explain, how you choosed the different sampling rates? Does the sampling rate depends on the used hardware or what is the reason for the different sampling rates?

    The sampling rate for the different vehicles was determined by the measurement equipment installed in each vehicle.

  • Line 175-...: The vehicles were tested on different routes. This is an good approach to get realisitic results. Nevertheless, how do you select these routes? Please explain the selection of the analyzed routes.

    The test routes represent a typical use case for each vehicle (heavy bus, heavy lorry and medium lorry). The shares of motorway, rural and urban of each route were derived from the standardized driving cycles in VECTO, used for the certification of the fuel consumption of new heavy duty vehicles. This has been clarified in the description of the test cycles:

    “All routes contain motorway, rural and urban driving and represent a typical use cases for each vehicle. Routes 2 and 4 have a higher share of highway driving, while Routes 3 and 5 have a higher share of urban and rural driving. The distance-based driving shares of each route are listed in Table 3 and were derived for each route from the mission profiles in VECTO, based on the vehicle category.”

  • Line 206: Why do you applied the filter in a window of 31 subsequent values and not 30 or 32 subsequent values?

    Regarding your question, the window must be a positive odd integer. We have added extra text as information:

    “This filter is applied in a window of 31 subsequent values. The window length value must be a positive odd integer [25], and is a compromise between the need for smoothing, and not distorting the original elevation shape.”

  • Line 225: Please check the blanks between "with" and "elevation".

    In the document downloaded from the MDPI page the blanks there seem corrected.

  • Equation 4: Can you please introduce abbreviations for the used paramaters, e.g. Fi_steady speed segment => Fi_sss. This only an suggestions, but the names of the parameters are very long.

    The authors prefer to avoid abbreviations in parameter names for the sake of clarity.

  • Line 258: Please check the blank between "in" and "[25]".

    Corrected as suggested.

  • Figure 4: Please start with an upper letter in the green boxes, compare e.g. "evelation data" with "Vehicle mass".

    Corrected as suggested.

  • Figure 4: The blue box "median wheel powers..." needs to be increased.

    Corrected.

  • Figure 4: "Correction of the side winds" => does this mean, that the wind from the front is not relevant?

    The yaw angle is the resultant of the side and head wind.

  • Line 275: What is the reason for the used correction value of 4%? Please explain, why you use 4%.

    The extra 4 % is derived by the default value proposed in the EU CO2 type approval certification procedure of LDVs [10], with an additional 1 % to account for the rotating mass of the wheel torque meters.
    This text is also added in the text:

    “Where F0raw, F1raw, F2raw are the raw road load coefficients derived by the fitting process, m is the vehicle mass with an addition of 4% to account for the rotational mass. The extra 4% is derived by the default value proposed in the EU CO2 type approval certification procedure of LDVs [11], with an additional 1% to account for the rotating mass of the wheel torque meters.”

  • Equations 6-8: Please use abbreviation for the parameters.

    Αn abbreviation would make the text slightly more comprehensive, still a non-widely used abbreviation might confuse the reader. For clarity, the authors prefer to avoid abbreviation in that point.

  • Equation 6: "step of 10 km/h" => Why?

    Added extra text above the equation to explain the origins of the practice followed in that sequence of equations:

    “It is a practice followed in the LDV EU CO2 certification procedure in the vehicle road load derivation with the coast down method. It is used to exclude the influence of the F1 factor when comparing two different vehicle configurations. This way, the difference in F0 can be attributed to the different tyre efficiency, and the difference in the F2 in the different Cd∙A.The coast down times are collected with a step of 10 km/h starting from maximum speed of 130 km/h. For HDV vehicle, as max value we use the more realistic 110 km/h.”

  • Line 286: "Road Load coefficients" vs. "road load coefficients". Please check the upper letters.

    Corrected all the occurrences where upper letter was used to lower letter in the whole text.

  • Line 293: It is similar to my comments above. You use some data or boundaries without an explanation. The missing explinations are the main reason for my assessment "Must be improved" regarding the "Are the methods adequately described".

Why 10 and 100 km/h?

We added text with the reasoning:

“The reason for this choice is explained in detail in [22] and has to do with the influence of the original F1raw contribution. Using the full range of reference speeds as in equation (6) for the linear fitting, the line produced by the adjusted road load coefficients (F1test, F1test, F2test) might not pass through the experimentally derived points for low and high speeds, affecting the calculation of RRC and Cd∙A, respectively.”

  • Line 315: "2 Hz". Why you do not use a greater sampling rate? Is the sampling rate limited by the use hardware?

    The sampling rate for the different vehicles was indeed determined by the measurement equipment installed in each vehicle.

  • Line 321: You stated that your results are demonstrating an accurate vehicle model. Why is it accurate? Is your method / model accurate because it is more precise compared to existing methods?

    The accuracy mentioned here relates to the accuracy of the vehicles' powertrain model in VECTO (drivetrain losses and engine maps) used to simulate the fuel consumption in section 3.4. This has been clarified in the manuscript. The powertrain model can be considered accurate for the purpose of this work with an error of less than 1.2%, as the simulations based on the road loads have a larger error of the fuel consumption: 1.54% - 3.94%, shown in Figure 13.

  • Figure 6: Please make lines in legend thicker => "Torque sensor 1" and "Torque sensor 2"

    Corrected as suggested.

  • Line 346-348: Please explain it in detail, why it is impossible to obtain observations in speeds lower than 35 km/h.

    The reason was the initial purpose of the tests.
    The road tests were conducted in framework of the regulatory vehicle testing procedure for heavy duty vehicles (EU 2019/318). The data analysis revealed that this test procedure does not contain steady speed segments at a sufficiently low speed to accurately determine the RRC with the proposed methodology. We added the text in the second paragraph in the `Results and Discussion` section:

    “The methodology proposed in the present study tries to identify both tire rolling resistance coefficient and air drag coefficient. However, the road tests were conducted in framework of the regulatory vehicle testing procedure for heavy duty vehicles [5]. Thus, due to the dynamicity of the test cycles, it was impossible to obtain observations in speeds lower than 35 km/h, where the impact of the air can be neglected.”

  • Line 354: Introduce the abbreviation "OEM".

    Introduced it in the very first time it is mentioned:

    “These boundaries were set for LDV’s, where the Original Equipment Manufacturer (OEM) official road load coefficients (F0OEM, F1OEM, F2OEM) were available and experimentally derived.”.

  • Line 357: Please do not use abbreviations in subtitles.

    We have corrected in all the subtitles mentioned in the results (3.1, 3.2, 3.3).

  • Line 379-381: You say "might be" and "can". How would you like to identify the real reason for the average error of -6.8%? May be your method is not suitable for the shown vehicles as well as routes. 

The hypothesis is validated by the simulations, where the fuel consumption is underestimated in general.

  • Line 416: "Figure 1" => "Figure 10"

    Corrected as suggested.

  • Line 425-426: Why 91 km/h and not 90 or 92 km/h etc.?

    The reason is that the for the specific constant speed segment, the average speed was 91 km/h. We have added this additional clarification in the text:

    “An example is shown in Figure 11 which takes into account the constant speed measurements with average speed of 91 km/h”.
  • Figure 12: Increase the label "error [%]" of the x-axis.

    Increased as suggested.

  • Table 8: Can you use abbreviations for the paramerters?

    The authors prefer to avoid abbreviations in parameter names for the sake of clarity.

  • Line 485-486: "4.3" and "%" are in two different lines.

    This was a result of having a blank between the number and the symbol ‘%’. We have taken out the blank from all the instances inside the text.

  • Line 527: Please check the blank between "." and "The".

    Corrected as suggested.

  • Line 627-634: Please do not use Wikipedia as a reference for the filter etc.

    We replaced all references to Wikipedia with the original sources.

  • In addition, the inegration of more literature (e.g. 3-4 paper) regarding common methods for the determination of aerodynamic and rolling restistances would be improve your paper. Based on these references, you are able to highligt the weakness of these methods and the strenghthes of your method.

Introduced 2 more studies.
One for coast down methodology [21]:

“In a study by the United States Environmental Protection Agency (US EPA) the methodology showed the ability to determine aerodynamic differences between tractors, and the effectiveness of devices to improve the aerodynamics. The standard error was below 2%, and the importance of at least 14 repetitions to reduce the uncertainty was pointed out [21].”

and one for constant speed tests [24]:

“The advantages of this method include the exclusion of the drivetrain losses and its high repeatability [23,24]. Regarding repeatability, in [24], US EPA performed in the same test site and same HDVs, both coast-down and constant speed test methodologies. The latter had lower standard deviation of the mean (under 0.5%), and lower dependency on the wind yaw angle.”

Their performances were highlighted, giving the opportunity for comparison to the methodology presented.
